# Reduced *Zeb1* Expression in Prostate Cancer Cells Leads to an Aggressive Partial-EMT Phenotype Associated with Altered Global Methylation Patterns

**DOI:** 10.3390/ijms222312840

**Published:** 2021-11-27

**Authors:** Jenna Kitz, Cory Lefebvre, Joselia Carlos, Lori E. Lowes, Alison L. Allan

**Affiliations:** 1London Regional Cancer Program, London Health Sciences Centre, Department of Anatomy & Cell Biology, Western University, London, ON N6A 5W9, Canada; jkitz@uwo.ca (J.K.); clefebvre2019@meds.uwo.ca (C.L.); 2Department of Medical Biophysics, Western University, London, ON N6A 5C1, Canada; jcarlos6@uwo.ca; 3Flow Cytometry, London Health Sciences Centre, London, ON N6A 5W9, Canada; lori.lowes@lhsc.on.ca; 4Department of Oncology, Western University, London, ON N6A 5W9, Canada; 5Cancer Research Laboratory Program, Lawson Health Research Institute, London, ON N6C 2R5, Canada

**Keywords:** prostate cancer, metastasis, epithelial-to-mesenchymal transition (EMT), partial-EMT (p-EMT), *Zeb1*, DNA methylation, 5-azacytidine

## Abstract

Prostate cancer is the most common cancer in American men and the second leading cause of cancer-related death. Most of these deaths are associated with metastasis, a process involving the epithelial-to-mesenchymal (EMT) transition. Furthermore, growing evidence suggests that partial-EMT (p-EMT) may lead to more aggressive disease than complete EMT. In this study, the EMT-inducing transcription factor *Zeb1* was knocked down in mesenchymal PC-3 prostate cancer cells (Zeb1^KD^) and resulting changes in cellular phenotype were assessed using protein and RNA analysis, invasion and migration assays, cell morphology assays, and DNA methylation chip analysis. Inducible knockdown of *Zeb1* resulted in a p-EMT phenotype including co-expression of epithelial and mesenchymal markers, a mixed epithelial/mesenchymal morphology, increased invasion and migration, and enhanced expression of p-EMT markers relative to PC-3 mesenchymal controls (*p* ≤ 0.05). Treatment of Zeb1^KD^ cells with the global de-methylating drug 5-azacytidine (5-aza) mitigated the observed aggressive p-EMT phenotype (*p* ≤ 0.05). DNA methylation chip analysis revealed 10 potential targets for identifying and/or targeting aggressive p-EMT prostate cancer in the future. These findings provide a framework to enhance prognostic and/or therapeutic options for aggressive prostate cancer in the future by identifying new p-EMT biomarkers to classify patients with aggressive disease who may benefit from 5-aza treatment.

## 1. Introduction

Prostate cancer is the second leading cause of cancer related deaths in American men [1]. Most of these deaths are caused by metastasis, which allows cancer to spread beyond the prostate to other parts of the body [2]. Metastasis is associated with an epithelial-to mesenchymal transition (EMT), where epithelial cells lose their epithelial characteristics and gain a mesenchymal phenotype, which aids in the process of metastasis [2,3,4,5,6,7,8].

Transcription factors bind to specific promoter sequences within the DNA to influence the expression of target genes [9]. Master EMT-inducing transcription factors upregulate mesenchymal genes and/or inhibit epithelial genes, which can cause the cell to undergo EMT [10]. An example of this is zinc finger E-box-binding homeobox 1 (Zeb1), which binds to the E-box promoter sequence, regulates neuronal differentiation, and has important roles in promoting EMT to allow for cell movement during gestation [11,12]. In cancer progression, Zeb1 promotes metastasis and a loss of cell polarity by repressing the epithelial proteins E-Cadherin and EpCAM and promotes tumorigenicity by repressing stemness-inhibiting microRNAs [10,13].

It is well-established that EMT is a dynamic state, utilizing both EMT and a reverse mesenchymal-to-epithelial (MET) transition to switch between epithelial and mesenchymal states during the process of metastasis [2,3,4,5,6,7,8]. In addition to EMT and MET, recent studies have demonstrated that there is an intermediate state called partial EMT (p-EMT), a phenotype that may result in the most aggressive cancer cells [14]. Partial EMT is associated with increased cell-cell interactions and cell proliferation in migrating circulating tumor cells (CTC). Growing evidence suggests that migrating cell clusters and CTC clusters in the blood are more aggressive and have higher metastatic potential than migrating single cells or single CTCs, and that these clusters often exhibit a p-EMT phenotype rather than complete EMT [15]. It has also been suggested that epigenetic modifications such as DNA methylation of the promoter region of essential genes may be responsible for this increased cell aggressiveness, and that treatment with a global de-methylating agent may aid in treatment of aggressive prostate cancers [16].

In the current study, we tested the hypothesis that knockdown of the EMT-inducing transcription factor *Zeb1* in mesenchymal PC-3 cells would produce an MET leading to a more epithelial, less aggressive phenotype compared to control cells. Unexpectedly, we observed that inducible knockdown of *Zeb1* in PC-3 cells (Zeb1^KD^ cells) resulted in a p-EMT phenotype including co-expression of epithelial and mesenchymal markers, a mixed epithelial/mesenchymal morphology, increased invasion and migration, and enhanced expression of p-EMT markers relative to PC-3 mesenchymal controls (ctrl cells). Treatment of Zeb1^KD^ cells with the global de-methylating drug 5-azacytidine (5-aza) [17] mitigated the observed aggressive p-EMT phenotype. DNA methylation chip analysis revealed 10 potential targets for identifying and/or targeting aggressive p-EMT prostate cancer in the future. These novel findings provide a framework to enhance prognostic and/or therapeutic options for aggressive prostate cancer in the future by identifying new p-EMT biomarkers to classify patients who may benefit from combination treatment with the clinically relevant inhibitor 5-azacitadine.

## 2. Results

### 2.1. Inducible Knockdown of Zeb1 in PC-3 Human Prostate Cancer Cells Results in Enhanced Expression of Epithelial Proteins

Mesenchymal human PC-3 prostate cancer cells were engineered with an inducible lentiviral shRNA system to knockdown expression of the master EMT regulator *Zeb1*. The following cell lines were created: PC-3 ctrl cells with a non-targeting control sequence of scrambled shRNA, and Zeb1^KD^ cells with shRNA targeting the 3′UTR of *Zeb1*. This was achieved using the SMARTvector inducible lentiviral shRNA (Dharmacon), which features Tet-on^®^ induction of the target shRNA in the presence of doxycycline (Dox) and validation by concurrent induction of TurboGFP (green fluorescent protein). Following Dox induction (72 h), we observed that Zeb1 protein (Figure 1A,B) and RNA (Appendix A) expression were significantly decreased compared to all ctrl cells (*p* ≤ 0.05), down to a level equivalent to that of human LNCaP cells, an epithelial prostate cancer cell line. Immunofluorescence confirmed successful knockdown of Zeb1 via TurboGFP expression following Dox induction (Appendix A). Immunoblotting (Figure 1C,D) and qRT-PCR (Appendix A) was used to assess EMT phenotypic marker expression following Dox induction of Zeb1^KD^ cells. Zeb1^KD^ cells had significantly higher expression of epithelial (EpCAM, E-Cadherin) proteins relative to ctrl cells (*p* ≤ 0.05), with no change in expression of mesenchymal proteins (Vimentin, N-Cadherin) (Figure 1C,D).

### 2.2. Knockdown of Zeb1 in PC-3 Prostate Cancer Cells Increases Migration and Invasion but Does Not Alter Proliferation

Next, we assessed the effect of *Zeb1* knockdown on migration and invasion of PC-3 prostate cancer cells using transwell migration (gelatin) and physical barrier wound healing assays. Unexpectedly, we observed that Zeb1^KD^ cells with Dox exhibit significantly increased migration compared to ctrl cells in both transwell (Figure 2A,B) and wound healing assays (Figure 2C,D) (*p* ≤ 0.05). When Zeb1^KD^ cells were assessed for changes in cell invasion using transwell invasion and spheroid invasion (Matrigel) assays, we similarly observed that Zeb1^KD^ cells with Dox demonstrate significantly enhanced invasion into Matrigel in both the transwell (Figure 3A,B) and spheroid invasion assays (Figure 3C,D) (*p* ≤ 0.05). BrdU proliferation assays were used to assess differences in cell proliferation between Zeb1^KD^ and ctrl cells, however no significant differences in proliferation were observed (Appendix A).

### 2.3. Knockdown of Zeb1 in PC-3 Prostate Cancer Cells Leads to a Partial EMT Phenotype at the Cellular and Molecular Level

We had originally expected that knockdown of *Zeb1* in mesenchymal PC-3 prostate cancer cells would lead to a mesenchymal-to-epithelial (MET) transition and reduced metastatic cell behaviors such as migration and invasion. Our observation that knockdown of *Zeb1* instead actually led to more aggressive cell behavior led us to investigate the potential for a partial EMT (p-EMT) phenotype [14]. Zeb1^KD^ cells with Dox were assessed for changes in cell morphology as described in the Materials & Methods section and in Appendix A. We observed that Zeb1^KD^ cells with Dox demonstrate a mixed cell morphology, with a significantly higher percentage of epithelial cells and significantly lower percentage of mesenchymal cells compared to ctrl cells (*p* ≤ 0.05) (Figure 4A,B). We next assessed changes in expression of the p-EMT markers P-Cadherin (P-Cad) and integrin β4 (ITGβ4) [18,19]. We observed that both P-Cad and ITGβ4 protein expression was significantly enhanced in Zeb1^KD^ cells with Dox compared to ctrl cells (*p* ≤ 0.05) (Figure 4C), while *P-Cad* RNA expression was also significantly increased in Zeb1^KD^ cells with Dox compared to ctrl cells (*p* ≤ 0.05) (Figure 4D).

### 2.4. Treatment of PC-3 Zeb1^KD^ Prostate Cancer Cells with the Global Demethylating Agent 5-Azacitadine Results in Decreased DNA Methylation, Migration, and Invasion

It has been suggested that epigenetic modifications such as DNA methylation of the promoter region of essential genes may be responsible for increased cell aggressiveness in cancer [16]. The global demethylating agent 5-aza is currently used to treat myelodysplastic syndrome [20] and is in many phase III clinical trials for cancer (ClinicalTrials.gov (accessed on 7 September 2021). To begin investigating whether DNA methylation is involved in the p-EMT phenotype observed in our Zeb1^KD^ cells, we treated cells with 5-aza ± Dox to assess the effects on cell phenotype. We observed that DNA methylation was decreased (based on decreased expression of 5-mC) in Zeb1^KD^ with Dox and ctrl cells treated with 5-aza compared to DMSO (*p* ≤ 0.05) (Figure 5A,B). We next assessed the effects of demethylation on cell aggressiveness, and observed that treatment with 5-aza significantly mitigated both migration (Figure 5C,D) and invasion (Figure 5E,F) compared to treatment with DMSO (*p* ≤ 0.05), although there was no change in proliferation (Appendix A).

### 2.5. Methylation Chip Analysis of Zeb1^KD^ PC-3 Prostate Cancer Cells Identified 10 Genes Associated with a p-EMT Phenotype

To explore specific molecular characteristics in Zeb1^KD^ cells that are being affected by demethylation, DNA was extracted from Dox-induced Zeb1^KD^ cells treated with DMSO (Z0) or 5 µM of 5-aza (Z5), and from Dox-induced ctrl cells treated with DMSO (C0) or 5-aza (C5) and assessed for global changes in DNA methylation using an Infinium MethylationEPIC chip. We observed over 100,000 differentially methylated sites between ctrl + DMSO cells (C0) and Zeb1^KD^ + DMSO cells (Z0) (false discovery rate (FDR) cutoff value = 0.05) (Figure 6A). We then further assessed only those sites which had an increase in DNA methylation between C0 and Z0 that also demonstrated rescued demethylation in Zeb1^KD^ cells + 5-aza (Z5); resulting in 51 potential sites of importance (FDR cutoff value = 0.05) (Figure 6B). Of these, 10 sites (*LRPPRC*, *CLDN11*, *MTOR*, *EPB41*, *DAPK1*, *PPZR2B*, *ZDHHC2*, *HSD17B13*, *MYOM2* and *MAN1A1*) were linked to decreased expression and increased aggressiveness/p-EMT, which may be of clinical importance for identifying an aggressive p-EMT phenotype in prostate cancer patients in the future (Figure 6C, Table 1).

### 2.6. MAN1A1, EPB41, HSD17B13 and MYOM2 Are Altered in Prostate Cancer Patients

Finally, we were interested in determining the potential clinical relevance of the identified DNA methylation targets in prostate cancer patients. We analyzed the 10 identified target p-EMT genes using available Ualcan (http://ualcan.path.uab.edu (accessed on 10 September 2021) and cBioportal (https://www.cbioportal.org (accessed on 10 September 2021) online clinical databases. We observed significant hypermethylation in 4 of the 10 target genes (*MAN1A1*, *EPB41*, *HSD17B13*, and *MYOM2*) in primary prostate cancer patient tumors (n = 503) compared to normal prostatic samples (*n* = 50) (*p* ≤ 0.05) (Figure 7A). Expression of *MAN1A1* was also observed to be significantly decreased in metastatic prostate cancer patients (*n* = 42) relative to non-metastatic prostate cancer patients (*n* = 44) (*p* ≤ 0.05) (Figure 7B). Lastly, we observed that decreased expression of *MYOM2* correlates with decreased overall survival in prostate cancer patients (Figure 7C). Taken together, these observations in prostate cancer patients support our pre-clinical findings in aggressive Zeb1^KD^ cells and suggest that these genes merit future investigation as potential biomarkers for combination treatment of prostate cancer patients with 5-aza.

## 3. Discussion

Prostate cancer is the most common cancer in American men and the second leading cause of cancer-related death. The majority of these deaths are associated with metastasis, a process involving the epithelial-to-mesenchymal (EMT) transition. Furthermore, growing evidence suggests that a partial-EMT (p-EMT) phenotype, whereby cells are able to simultaneously maintain both epithelial and mesenchymal characteristics, may lead to more aggressive disease than complete EMT [14]. Gaining a greater understanding of p-EMT may thus provide insights into the mechanisms of metastatic disease progression, which currently has no cure. In the current study, we observed that inducible knockdown of *Zeb1* in mesenchymal PC-3 cells resulted in a p-EMT phenotype including co-expression of epithelial and mesenchymal markers, a mixed epithelial/mesenchymal morphology, increased invasion and migration, and enhanced expression of p-EMT markers.

In addition to changes in gene and protein expression, the p-EMT phenotype is commonly associated with aberrant hypermethylation [31,32]. The global de-methylating agent 5-azacytidine (5-aza) is FDA-approved for treating myelodysplastic syndrome and is currently in 42 phase III clinical trials for treating cancer patients (ClinincalTrials.gov (accessed on 7 September 2021), as well as 4 phase II clinical trials specifically for prostate cancer patients (ClinicalTrials.gov (accessed on 7 September 2021). When we treated our p-EMT prostate cancer cells with 5-aza, we observed a significant decrease in aggressive phenotype. Furthermore, our DNA methylation chip analysis revealed 10 potential markers for further investigation in association with p-EMT.

Our observations included increased DNA methylation of *EPB41* and *HSD17B13*. *EPB41* has been identified as a tumor suppressor in the molecular pathogenesis of meningiomas [24]. *HSD17B13* expression has also been shown to inhibit the progression and recurrence of hepatocellular carcinomas [28]. Additionally, Ualcan online database analysis showed increased promoter methylation of both *EPB41* and *HSD17B13* in prostate cancer patients compared to healthy controls. Silencing of these genes due to increased DNA methylation could result in tumor progression and poor patient survival [24,28].

We also observed increased DNA methylation of *MAN1A1*, which correlated with decreased gene expression. Reduced *MAN1A1* expression has previously been associated with reduced survival in breast cancer patients [30]. In our study, Ualcan online database analysis showed increased promoter methylation of *MAN1A1* in prostate cancer patients compared to healthy controls and in metastatic prostate cancer patients compared to non-metastatic prostate cancer patients. This suggests that decreased expression of *MAN1A1* may be associated with increased prostate cancer aggressiveness and could be a novel marker for identifying a p-EMT phenotype in patient tumors.

Lastly, we demonstrated increased DNA methylation of *MYOM2*. *MYOM2* has been previously been observed to be downregulated in breast cancer patients, as determined by multiplex RT-PCR [29]. Our assessment using the cBioportal online database revealed that decreased *MYOM2* expression is associated with significantly worse progression free survival in prostate cancer patients compared to those with high *MYOM2* expression, suggesting that *MYOM2* may be another potential marker for identifying aggressive prostate cancer.

In summary, in this study we developed a stable, inducible p-EMT prostate cancer model that provides the opportunity to investigate the aggressive p-EMT phenotype, a cell state that often occurs transiently in vivo. In addition, we have identified 4 potential biomarkers related to p-EMT for which decreased expression may be an indicator of metastatic disease and may warrant consideration for use in identifying patients who would benefit from 5-aza treatment to target hypermethylation. Currently, there is no cure for metastatic prostate cancer, however, early detection and targeted treatment with agents that target hypermethylation may slow down the progression towards metastasis and improved patient outcomes.

## 4. Materials and Methods

### 4.1. Cell Culture

Human mesenchymal PC-3 prostate cancer cells (parental PC-3 cells [#CRL-1435]; ATCC, Manassas, VA, USA) were cultured in F12K media + 10% fetal bovine serum (FBS). Human epithelial LNCaP prostate cancer cells (#CRL-1740, ATCC) were cultured in RPMI-1640 media + 10% FBS. Human epithelial MDA-MB-468 breast cancer cells (#HTB-132, ATCC) were cultured in alpha minimum essential media (αMEM) + 10% FBS. Cell lines were authenticated via third party testing (IDEXX BioAnalytics, Columbia, MO, USA). Primary lung fibroblasts (Lonza, Basel, Switzerland) were cultured in RPMI-1640 media + 5% FBS, 1% 4-(2-hydroxyethyl)-1-piperazineethanesulfonic acid (HEPES), 0.1% bovine serum albumin (BSA) (10%), 0.5% insulin, and 0.05% hydrocortisone. Media and reagents are from Life Technologies (Carlsbad, CA, USA), and FBS is from Sigma (St. Louis, MO, USA).

### 4.2. Cell Transductions

To create PC-3 Zeb1^KD^ and ctrl cells, 1 × 10^6^ PC-3 cells/mL were seeded into each well of a 6-well dish 24 h prior to transduction. Twenty-five μL of SMARTvector Lentiviral *Zeb1* shRNA stock (target region; 3′ untranslated region, target sequence 5′-TCTAAACCCAGGCTTCCCT-3′) or scrambled control (non-targeting control sequence) (Dharmacon, Lafayette, CO, USA) was added to each well and growth media was exchanged for transduction media containing 0.01% polybrene. After 24 h, transduction media was exchanged for growth media. One day later, growth media was exchanged for selection media containing 0.025% puromycin. Cells were then cultured as usual, supplementing growth media with 0.025% puromycin to continue selective pressure. Resulting changes in inducible Zeb1 expression (± Dox) were analyzed using immunoblotting and qRT-PCR as described below.

### 4.3. Immunoblotting

Cells were harvested by cell scraping, collected in lysis buffer, and quantified using a Lowry Assay. Protein (10 μg) was subjected to sodium dodecyl sulfate polyacrylamide gel electrophoresis (SDS-PAGE) and transferred onto polyvinylidene difluoride membranes (PVDF; Millipore, Billerica, MA, USA). Membranes were blocked using 5% bovine serum albumin (BSA) in Tris-buffered saline + 0.1% Tween-20 (TBS-T). Anti-human primary antibodies were diluted in 5% BSA in TBS-T prior to use as detailed in Appendix A. Goat anti-mouse IgG and goat anti-rabbit IgG secondary antibodies (Calbiochem, Billerica, MA, USA) conjugated to horseradish peroxidase and diluted in 5% BSA/TBS-T were used at concentrations of 1:2000 and 1:5000. Protein expression was visualized using Amersham ECL Prime Detection Reagent (GE Healthcase, Wauwatosa, WI, USA), and normalized to total protein based on amido black (Sigma) staining of membranes or actin immunoblotting.

### 4.4. Quantitative Real-Time PCR

Total RNA was isolated using TRIzol (Life Technologies), and reverse transcribed using SuperScript™ IV VILO Master Mix (Invitrogen, Waltham, MA, USA; 11766050). Samples were then subjected to subsequent RNA analysis using Advanced qPCR Master Mix with Supergreen LO-ROX (Wisent Bioproducts, Saint-Jean-Baptiste, QC, Canada) on a QuantStudio™ 3 Real-Time PCR system (Applied Biosystems, Waltham, WA, USA) with primers detailed in Appendix A. *GAPDH* was used as a control.

### 4.5. Transwell Migration and Invasion Assays

Changes in cell migration and invasion were assessed using transwell migration and invasion assays. Transwell plates were coated with either gelatin (4 µg/well, migration) or Matrigel (6 µg/well, invasion). Media in the bottom well included normal media supplemented with puromycin and 2% FBS (migration) or 5% FBS (invasion) with or without 1 µg/mL Dox treatment as required. Human PC-3 prostate cancer cells (parental, ctrl or Zeb1^KD^; 5 × 10^4^ cells/mL) were seeded onto the top portion of each transwell chamber and incubated for 18 h at 37 °C, 5% CO_2_ prior to staining and assessment of differences in migration and invasion. Five high powered fields of view (HP-FOVs) were captured for each well, and the mean number of migrated or invaded cells/HP-FOV was calculated using ImageJ software (National Health Institute, Bethesda, MD, USA).

### 4.6. Physical Barrier Wound Healing Assay

Changes in migratory capacity were also assessed using physical barrier wound healing assays. Cells (3 × 10^5^/mL) were plated in F12K media supplemented with puromycin and doxycycline, DMSO, and/or 5-aza, onto 24 well plates. Cells were incubated at 37 °C, 5% CO_2_. After 24 h the physical barrier was removed from each well. Images were captured at 0, 12, 24, and 36 h time points using 5 HP-FOVs for each well. Cell migration, calculated by percent wound closure, was analyzed using ImageJ. software.

### 4.7. Spheroid Invasion Assay

Changes in invasion were also assessed using spheroid invasion assays. Cells (5 × 10^3^) were plated onto 96-well ultra-low attachment plates spheroid microplates (Corning, Kennebunk, ME, USA) using growth media supplemented with puromycin, doxycycline, DMSO, and/or 5-aza and allowed to grow into spheroids for 96 h. Matrigel was added to the spheroids and images were captured at 0, 24, and 48 h time points using 5 HP-FOVs for each well. ImageJ software was used to calculate the area of invasion from spheroids into surrounding Matrigel.

### 4.8. BrdU Proliferation Assay

Cell proliferation was assessed using a bromodeoxyuridine (BrdU) incorporation assay. Cells were plated on 8-well chamber slides, allowed to adhere, and serum-starved for 72 h. Media was then replaced with F12K supplemented with puromycin and 10% FBS ± Dox, 5-aza, and/or DMSO for 24 h. Following incubation, Cell Proliferation Labelling Reagent (BrdU) (GE Healthcare, Chicago, IL, USA) was added for 30 min, cells were formalin fixed and stained with a 100 µL/well anti-BrdU primary antibody (BD-347580) and a 1:400 concentration of a PE-conjugated goat anti-mouse IgG secondary antibody was used for immunofluorescent visualization. Images were captured using 5 HP-FOVs for each well, and nuclei were counted using ImageJ, with results expressed as a percentage of BrdU positive cells to total nuclei (DAPI^+^).

### 4.9. Cell Morphology Assay

Changes in cell morphology were determined by analyzing the roundness versus spindle-like shape of each cell. High powered FOVs were used to capture cell images, and 250 cells per HP-FOV (n = 3) were analyzed for cell shape. The actual area (AA) of each cell was calculating by outlining and measuring the entire cell in ImageJ, which was also used to trace the diameter between the longest two points of each cell and the expected area (EA) was calculating using the equation *πr*^2^. The AA was then divided by EA to assign each cell with a number from 0 to 1. If AA was equal to the expected area then the number is 1, and the cell is more round in shape. If the AA is less than the expected area then the number is closer to 0, and the cell is more spindle shaped. To determine the limits of what number represented a round or spindle-shaped cell control, epithelial MDA-MB-468 breast cancer cells and mesenchymal primary lung fibroblasts were used as controls for cell shape (250 cells/FOV, n = 3). The average of the epithelial/mesenchymal control cells was attained, and the standard deviation was either added or subtracted from the average respectively in order to create a cutoff point for an epithelial cell, a mesenchymal cell, and a cell of “mixed” morphology (i.e., neither epithelial nor mesenchymal) (Appendix A).

### 4.10. DNA Extraction and Dot-Blot DNA Analysis

DNA was extracted using a Blood & Cell Culture DNA Mini Kit (Qiagen, Hilden, Germany) and the manufacturer’s protocol. For the dot-blot analysis, 180 ng of DNA was added to 3 M NaOH and incubated at 42 °C for 12 min to denature the DNA. Samples were immediately transferred to positively charged nylon membranes (Roche, Mannheim, Germany) in the dot-blot apparatus. Membranes were then baked at 120 °C for 30 min to allow DNA-membrane crosslinking. The membrane was then blocked in 1×TBS + 0.05% Tween-20 and 5% powdered milk for 1 h prior to incubation with the anti-5mC primary antibody (ab179898; 1:500 in blocking solution) and agitated for 1.5 h. Membranes were washed 3× with TBS-T for 10 min, and then incubated with a goat anti-mouse IgG secondary antibody (Calbiochem, Billerica, MA, USA; 1:1000) for 1 h. Expression of 5 mC was visualized using Amersham ECL Prime Detection Reagent (GE Healthcare) on a ChemiDoc™ MP Imaging System, and normalized to total DNA based on methylene blue staining of membranes.

### 4.11. DNA Methylation Chip Analysis

Changes in global DNA methylation profile were analyzed using the Illumina Methylation EPIC BeadChip (Illumina, San Diego, CA, USA) and 1000 ng of DNA input (n = 4 per cell group) using the manufacturer’s protocol [33]. In total, 3 different QC methods were carried out. First, raw methylation betas were generated using the Minfi package in R [34] and no QC was performed in order to retain flexibility for analysis. Secondly, quality control was performed with the Chip Analysis Methylation Pipeline (ChAMP) [35]. This method filtered probes with a detection *p*-value above 0.01 (removing 3337 probes), bead count <3 in at least 5% of samples (removing 26,519 probes), only keeping CpG methylation measurements (removing 2931 probes), filtering probes with SNPs (removing 95,596 probes), probes that align to multiple locations (removing 11 probes), filtering XY chromosome probes (removing 16,109 probes). The last method of QC still used ChAMP, but only removed probes failing detection *p*-value, bead count and non-cpg sites (as explained above). After probe filtering with ChAMP, no samples were removed due to QC issues, and values for each sample were normalized with BMIQ normalization [36].

### 4.12. Patient Sample Analysis

Follow-up analysis was completed using Ualcan and cBioportal online clinical patient databases. Using the gene analysis Ulcan database (accessed on 10 September 2021), each aberrantly methylated gene identified was analyzed. Utilizing the TCGA dataset, genes were assessed for promoter methylation in prostate adenocarcinoma compared to normal tissue as well as for expression in metastatic prostate cancer (MET500 dataset) compared to non-metastatic prostate cancer. Additionally, cBioportal (accessed on 10 September 2021) was used to assess for association with survival using mRNA expression level comparisons of aberrantly methylated genes in prostate adenocarcinoma. First, the sample set was identified using Onco Query Language on cBioportal (accessed on 10 September 2021). Patients were stratified based on expression of each identified gene, an mRNA profile was added to the query, and “example gene: EXP>2 EXP<-2” was written in the gene set box. After running the query, the “samples affected” list was downloaded. Next the list of sample IDs was pasted into the homepage into the “user-defined case list” in the “select patient/case set”: dropdown. This query only looks at samples with high or low expression. To stratify into high versus low survival analysis, “example gene: EXP>2” was entered in the gene set box and the same (prostate adenocarcinoma) mRNA profile was selected. The query was run, and the survival tab was selected for results.

### 4.13. Statistical Analysis

Statistical analysis was performed using GraphPad Prism 9 (GraphPad, San Diego, CA, USA) and Microsoft Excel 16.5.2 (Microsoft, Redmond, WA, USA). Unless otherwise stated, data is presented as the mean ± standard error of the mean (SEM), with *p* ≤ 0.05 considered to be statistically significant. For normally distributed comparisons of 2 groups, *t*-tests were performed and for comparisons of more than 2 groups a one-way ANOVA with follow up *t*-tests for multiple comparisons was performed. Non-matched, non-parametric data of more than two groups was assessed with a one-way Krustral-Wallis ANOVA with follow up Mann-Whitney tests for multiple comparisons, with a false discovery rate cutoff = 0.05 considered to be statistically significant.

## 5. Conclusions

In the current manuscript we created and characterized a stable inducible p-EMT cell line model by decreasing *Zeb1* expression in mesenchymal PC-3 prostate cancer cells. This resulted in an increased aggressive phenotype compared to mesenchymal controls. We identified 10 potential p-EMT markers which had aberrant DNA methylation in these p-EMT cells which may be used as a screening panel for p-EMT patients in the future to allow for earlier detection of aggressive prostate cancer and/or potentially serve to identify patients who might benefit from 5-aza therapy.

## Figures and Tables

**Figure 1 ijms-22-12840-f001:**
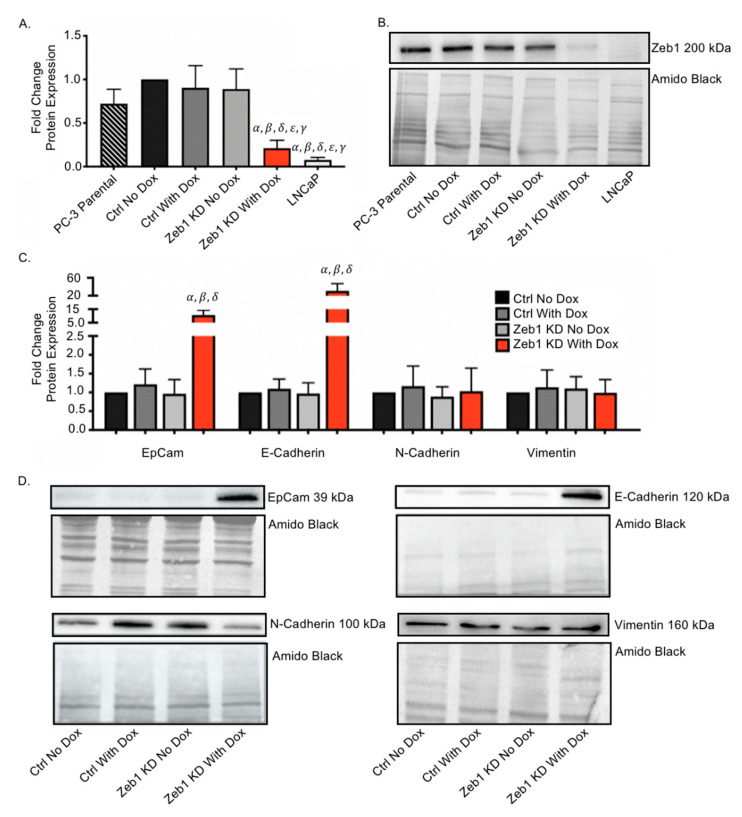
Inducible knockdown of *Zeb1* in PC-3 human prostate cancer cells results in enhanced expression of epithelial proteins. Mesenchymal human PC-3 prostate cancer cells were engineered to knockdown expression of the master epithelial-to-mesenchymal (EMT) regulator *Zeb1* using the SMARTvector inducible lentiviral shRNA system (Dharmacon), which features Tet-on^®^ induction of the target shRNA in the presence of doxycycline (Dox). (**A**,**B**) Immunoblot analysis of Zeb1 protein expression in the presence or absence of Dox (72 h) in Zeb1^KD^ (*Zeb1* knockdown), control (ctrl) PC-3 cells, or LNCaP cells. (**C**,**D**) Immunoblot analysis of E-Cadherin, EpCAM, Vimentin and N-cadherin in Zeb1^KD^ or ctrl cells 72 h after Dox induction. Representative immunoblots are shown and amido black staining of total protein was used as a loading control. Quantitative data is presented as mean ± standard error of the mean (SEM) fold-change in expression relative to ctrl cells (*n* = 3). *α* = significantly different than ctrl no Dox. *β* = significantly different than ctrl with Dox. *δ* = significantly different than Zeb1^KD^ no Dox. *γ* = significantly different than PC-3 parental *ε* = significantly different than LNCaP (*p* ≤ 0.05).

**Figure 2 ijms-22-12840-f002:**
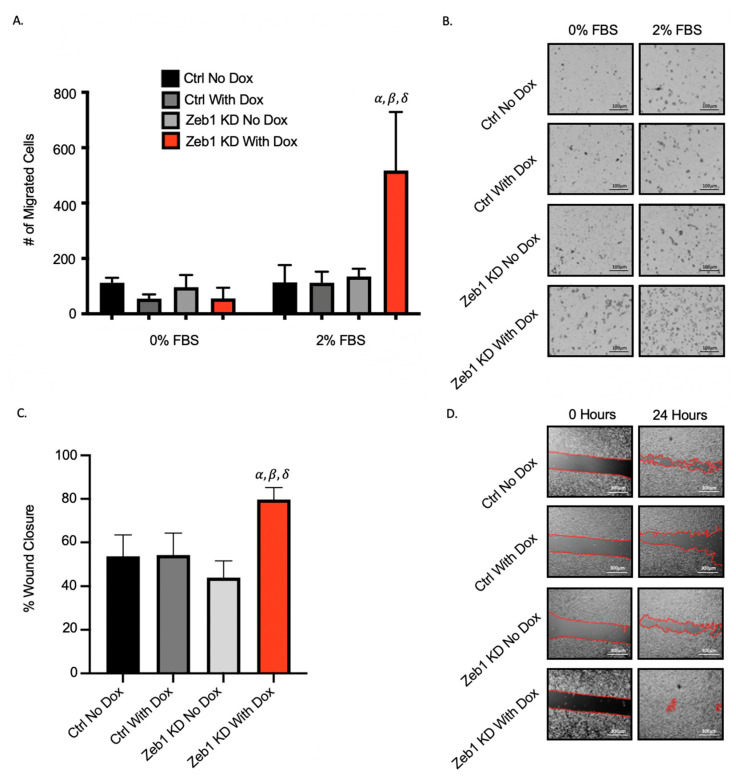
Knockdown of *Zeb1* in PC-3 prostate cancer cells increases cell migration. (**A**,**B**) Transwells were coated with 6 µg/well of gelatin. Cells (5 × 10^4^/well) were added to wells and either control media (0% fetal bovine serum [FBS]) or chemoattractant media (2% FBS) was added and cells were allowed to migrate for 18 h. Cells were fixed with 1% glutaraldehyde and mounted with DAPI-containing mounting media. (**C**,**D**) For physical barrier wound healing assays, cells were seeded and grown to 90–100% confluency. The physical barrier was removed and cells were allowed to migrate into the wound for 36 h. Representative images are shown for each assay; with migration calculated based on 5 high-powered fields of view (HP-FOV) per well. Black scale bars = 100µm, white scale bars = 300µm. Data is presented as the mean ± standard error of the mean (SEM) (*n* = 3). *α* = significantly different than control (ctrl) no doxycycline (Dox). *β* = significantly different than ctrl with Dox. *δ* = significantly different than Zeb1^KD^ (*Zeb1* knockdown) no Dox (*p* ≤ 0.05).

**Figure 3 ijms-22-12840-f003:**
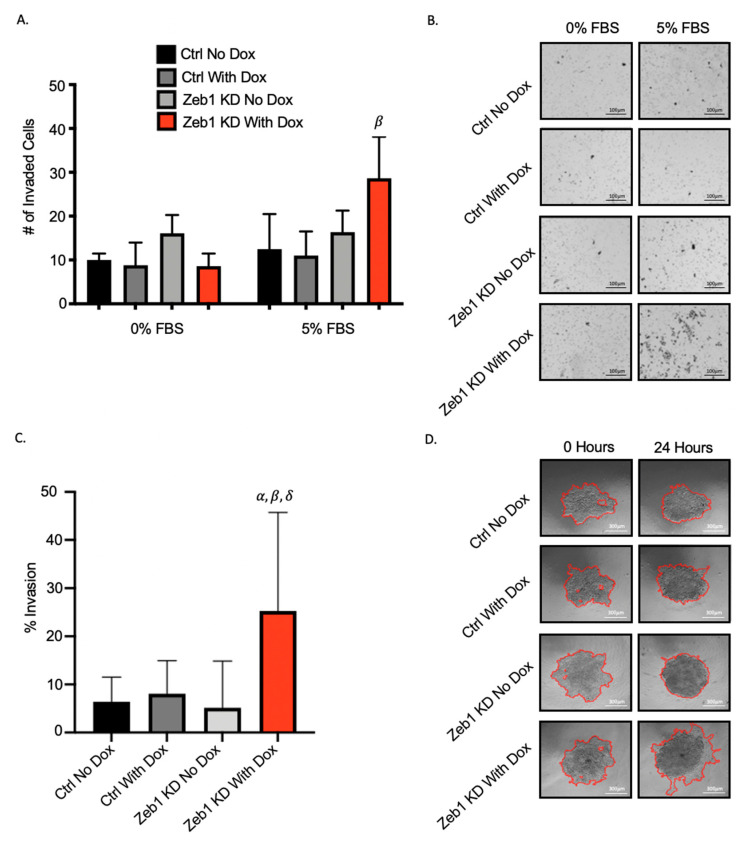
Knockdown of *Zeb1* in PC-3 prostate cancer cells increases cell invasion. (**A**,**B**) Transwells were coated with 4 µg/well of Matrigel. Cells (5 × 10^4^/well) were added to wells and either control media (0% fetal bovine serum [FBS]) or chemoattractant media (5% FBS) was added and cells were allowed to invade for 24 h. Cells were fixed with 1% glutaraldehyde and mounted with DAPI-containing mounting media. (**C**,**D**) For spheroid invasion assays, cells were seeded onto ultra-low attachment plates and allowed to grow for 96 h to create spheroids. Matrigel was then added and invasion was quantified after 48 h. Representative images are shown for each assay; with invasion calculated based on 5 high-powered fields of view (HP-FOV) per well. Black scale bars = 100 µm, white scale bars = 300 µm. Data is presented as the mean ± standard error of the mean (SEM) (*n* = 3). *α* = significantly different than PC-3 control (ctrl) no doxycycline (Dox). *β* = significantly different than ctrl with Dox. *δ* = significantly different than Zeb1^KD^ (*Zeb1* knockdown) no Dox (*p* ≤ 0.05).

**Figure 4 ijms-22-12840-f004:**
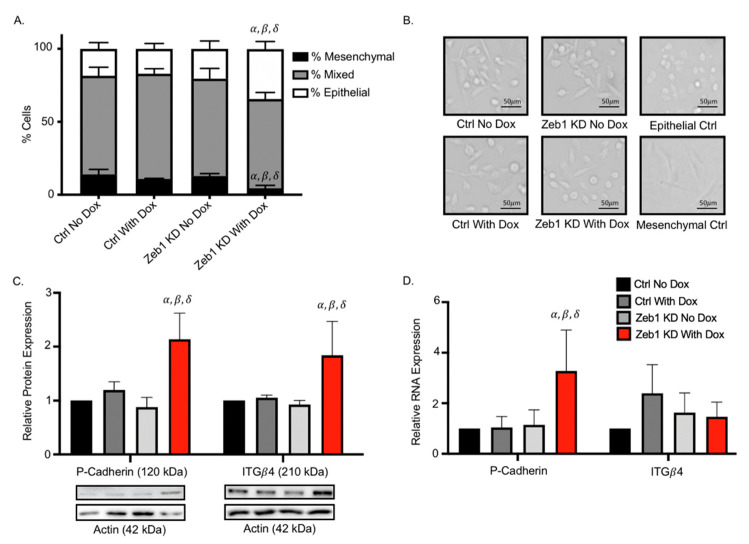
Knockdown of *Zeb1* in PC-3 prostate cancer cells leads to a partial-EMT phenotype at the cellular and molecular level. (**A**,**B**) Cultured PC-3 Zeb1^KD^ (Zeb1 knockdown) and control (ctrl) cells were assessed for cell morphology characteristics as described in the Materials & Methods and in Appendix A (N = 3; n = 250/cells per group). Representative images of each cell group and epithelial (MDA-MB-468) and mesenchymal (primary lung fibroblasts) controls are shown. (**C**) Immunoblot analysis of P-Cadherin and ITGβ4 in Zeb1^KD^ or ctrl cells. Actin was used as a loading control and representative immunoblots are shown. (**D**) qRT-PCR analysis of p-EMT marker expression in the presence of absence of Dox in Zeb1^KD^ or ctrl cells. Data is presented as the mean ± standard error of the mean (SEM) (n = 3) relative to ctrl no Dox. Scale bars = 50 µm. *α* = significantly different than PC-3 ctrl no Dox. *β* = significantly different than ctrl with Dox. *δ* = significantly different than Zeb1^KD^ no Dox (*p* ≤ 0.05).

**Figure 5 ijms-22-12840-f005:**
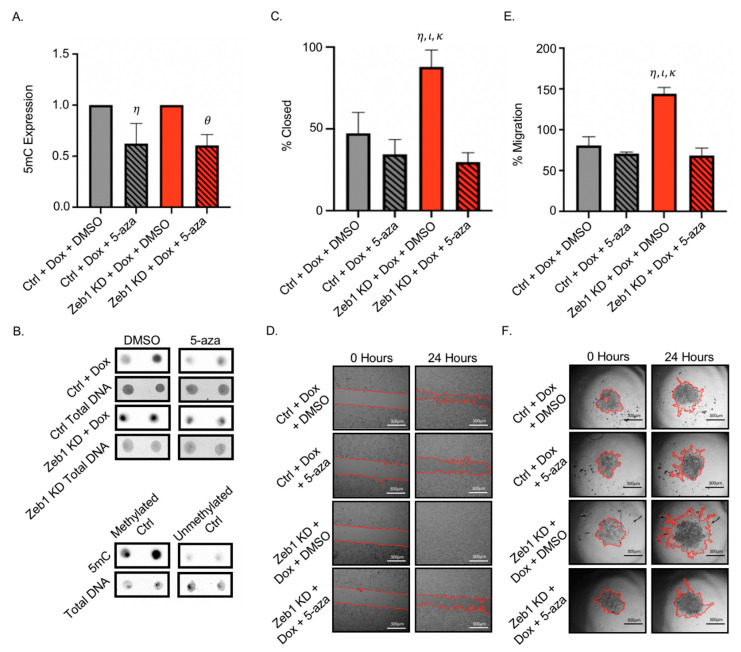
Treatment of PC-3 Zeb1^KD^ prostate cancer cells with the global demethylating agent 5-azacitadine (5-aza) results in decreased DNA methylation, migration and invasion. (**A**,**B**) PC-3 Zeb1^KD^ (*Zeb1* knockdown) with doxycycline (Dox) or control (ctrl) cells were treated with either dimethyl sulfoxide (DMSO) or 5-aza (5 µM) for 24 h and DNA was extracted to assess for global DNA methylation via dot blot assays. Representative dot blots are shown. Methylated and unmethylated DNA controls were used to validate 5-methylcytosine (5mC) expression. (**C**,**D**) Cells were seeded onto physical barrier cell culture dishes and grown to 90–100% confluency. Treatments (5 µM 5-aza or DMSO) were added to cells, the physical barrier was removed, and cells were allowed to migrate into the wound. (**E**,**F**) Cells were seeded onto ultra-low attachment plates and allowed to grow for 96 h to create spheroids. After 96 h of growth, Matrigel and 5 µM 5-aza or DMSO were added. Representative images are shown for each assay; with migration or invasion calculated based on 5 high-powered fields of view (HP-FOV) per well. Scale bars = 300 µm. Data is presented as the mean ± standard error of the mean (SEM) (n = 3). *η* = significantly different than ctrl with Dox and treated with DMSO. *θ* = significantly different than Zeb1^KD^ with Dox and treated with DMSO. *ι* = significantly different than ctrl with Dox treated with 5 µM 5-aza. *κ* = significantly different than Zeb1^KD^ with Dox treated with 5 µM 5-aza (*p* ≤ 0.05).

**Figure 6 ijms-22-12840-f006:**
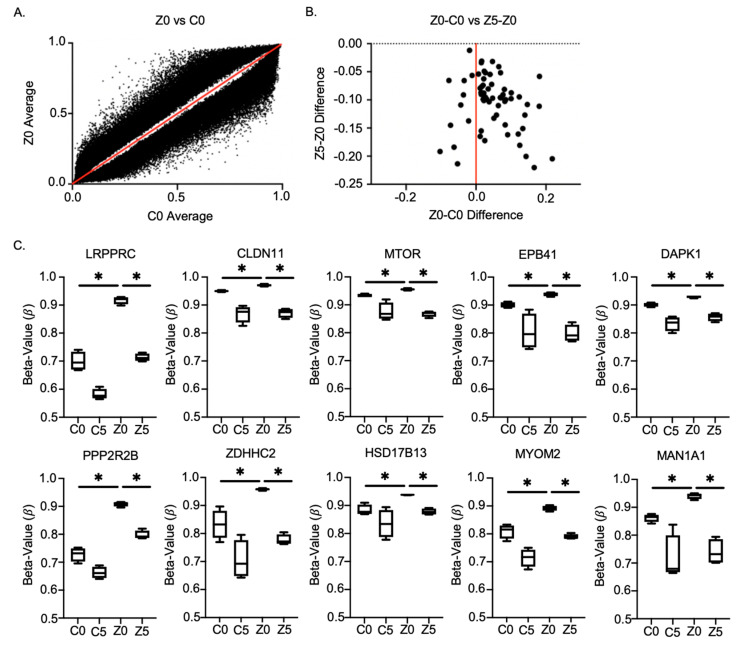
DNA methylation chip analysis of Zeb1^KD^ PC-3 prostate cancer cells identified 10 genes associated with a p-EMT phenotype. DNA was extracted from PC-3 Zeb1^KD^ (Zeb1 knockdown) cells with doxycycline (Dox) treated with dimethyl sulfoxide (DMSO) (Z0) or 5-azacitadine (5-aza; Z5; 5µM), and from Dox-induced control (ctrl) cells treated with DMSO (C0) or 5-aza (C5; 5 µM) and was assessed for global changes in DNA methylation using an Infinium Methylation EPIC chip. (**A**) A two-tailed, unpaired, equal variance *t*-test was completed with FDR cut-off value = 0.05 (Benjamini-Hochberg FDR) between C0 and Z0. This was filtered for significant Z0-C0 differences, and 107,971 cg sites were observed. (**B**) A two-tailed, unpaired, equal variance *t*-test was completed with FDR cut-off value = 0.05 (Benjamini-Hochberg FDR) between Z0 vs. Z5. This was filtered for significant Z0-Z5 differences, and 62 cg sites were observed. Among the C0-Z0 and Z0-Z5 significant differences, we wanted to identify rescue changes, so we filtered the dataset for cg sites where Z0-C0 = -(Z5-Z0) and identified 51 cg sites (right side of graph (**B**)). (**C**) Genes identified in DNA methylation chip analysis (increased DNA methylation from C0 versus Z0 with a corresponding demethylation in Z5). *β*-value represents the estimate of DNA methylation level at a given locus. Data is presented as the mean ± standard error of the mean (SEM) (n = 4). * = significant difference between conditions.

**Figure 7 ijms-22-12840-f007:**
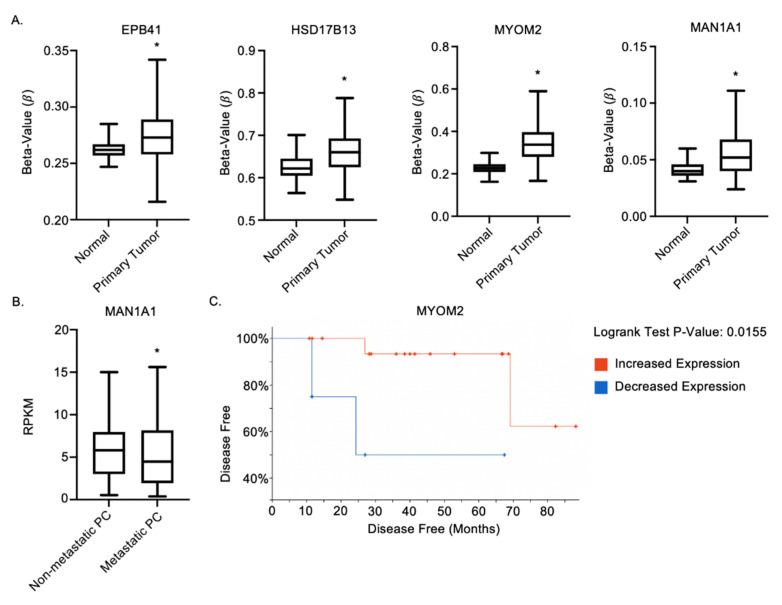
*MAN1A1*, *EPB41*, *HSD17B13* and *MYOM2* are altered in prostate cancer patients. (**A**,**B**) Ualcan analysis in prostate adenocarcinoma identified (**A**) 4 target genes (*MAN1A1*, *EPB41*, *HSD17B13* and *MYOM2*) with increased promoter methylation in primary prostate cancer tumors (n = 502) compared to normal prostatic samples (n = 50) and (**B**) *MAN1A1* RNA expression in metastatic prostate cancer (PC) (n = 42) vs. non-metastatic prostate cancer (n = 44). (**C**) cBioportal analysis of relationship between *MYOM2* expression and progression free survival. * = significantly different between conditions.

**Table 1 ijms-22-12840-t001:** Functional relevance of genes identified in DNA methylation chip analysis.

Gene	Function Relative to Cancer Aggressiveness
*LRPPRC*	Dysregulation is related to various diseases ranging from tumors to viral infections [21].
*Claudin-11*	Plays an important role in cellular proliferation and migration [22].
*mTOR*	Regulates cell growth, proliferation, motility, survival, protein synthesis, autophagy, and transcription [23].
*EPB41*	Expression is significantly decreased in HCC tissue specimens, especially in portal vein metastasis or intrahepatic metastasis, compared to normal tissues [24].
*DAPK1*	Downregulation promotes the stemness of cancer stem cells and EMT process by activating ZEB1 in colorectal cancer [25].
*PPP2RR2B*	Negative control of cell growth and division [26].
*ZDHHC2*	Tumor suppressor in metastasis and recurrence of HCC [27].
*HSD17B13*	Downregulated in hepatocellular carcinoma [28].
*MYOM2*	Downregulation was observed in a clinical assessment of breast cancer patients [29].
*MAN1A1*	Reduced expression leads to impaired survival in breast cancer [30].

## Data Availability

Infinium Methylation EPIC chip dataset can be found archived in the GEO repository, series number GSE186782, “Treatment of p-EMT prostate cancer cell with the demethylating drug 5-azacytidine reduces cell aggressiveness and changes methylation profile”.

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
