# Peer review of "Reduced Zeb1 Expression in Prostate Cancer Cells Leads to an Aggressive Partial-EMT Phenotype Associated with Altered Global Methylation Patterns"

_ijms, 2021, doi:10.3390/ijms222312840_

Round 1
Reviewer 1 Report
The work from Kitz J et al investigates the effects of the inducible knock down of the transcription factor Zeb1 in mesenchymal PC-3 prostate cancer cells. These cells, named Zeb1KD, presented an intermediate phenotype called “partial EMT”, p-MET, and they also showed co-espression of epithelial and mesenchymal markers, increased invasion and migration, and enhanced expression of p-EMT markers relative to PC-3 mesenchymal controls. Incubation of these cells with the de-methylating drug 5-azacytidine reduce the aggressiveness of the phenotype observed. Finally, the authors conducted a methylation chip analysis and identified 10 potential targets that can be used to enhance prognostic and/or therapeutic for aggressive prostate cancer in the future.
Although the work is potentially interesting, there are some points that needed to be improved:
- In the first paragraph, line 59, I would suggest to eliminate the term “silencing” after methylation (it is indicated in brackets ). Sometimes DNA methylation is associated to enhanced gene transcription ( Harris C. J et al Science 2018 Dec 7; 362 (6419):1182-1186).
- Line 58. Please, specify that you are talking about DNA methylation, since also histones could be methylated. Do the same correction through the whole text.
- Lines 73-74. The sentence “by identifying new p-EMT biomarkers to identify patients….”, should be corrected using a synonymous word.
- Line 88. The sentence “down to a level….”, should be corrected In “downing to a level….”
- Figure 1 (A, C). Please indicate standard error also for ctrl cells, since the experiments were done in triplicate
- Figure 1. Why do authors not use an housekeeping gene to normalize Immunoblots? In Figure 4 they show immunoblots in which they have used Actin as a loading control. I would suggest to do the same for experiments in Figure 1.
- Figure 2 and Figure 3. Authors should indicate which sample have used to calculate the fold change and put this value as 1.
- Figure 4 (C, D). Please, indicate the colors of bars which samples they represent
Reviewer 2 Report
In this study, Kitz and colleagues investigated the role of the EMT-inducing transcription factor Zeb1 in mesenchymal PC-3 prostate cancer cells. They found that Knock down of Zeb1 (Zeb1KD) resulted in a p-EMT phenotype with coexpression of epithelial and mesenchymal markers. Treatment of Zeb1KD cells with 5-azacytidine mitigated the p-EMT phenotype providing a new prospective in using 5-azacytidine treatment in specific patients.
The comment to the manuscript are reported below:
- In the text the authors refer to the cell line used in this study as PC3, however they used the PC3-EMT (#CRL-3471, ATCC). It should be corrected because it can be misunderstood.
- Why the authors did not compared Zeb1 expression with the PC3-epi subline (#CRL-3470, ATCC) instead of the parental PC3?
- How the authors normalized the bands of western blot? they used amido black but did they normalize using the whole lane or specific bands?
- in the materials and methods section it is not reported the parental PC3 cell line used. It must be added. Moreover, the authors should report the specific code number of the cell lines used in this study.
